# In Vitro and In Vivo Efficacy of a Stroma-Targeted, Tumor Microenvironment Responsive Oncolytic Adenovirus in Different Preclinical Models of Cancer

**DOI:** 10.3390/ijms24129992

**Published:** 2023-06-10

**Authors:** Ana Alfano, Eduardo G. A. Cafferata, Mariela Gangemi, Alejandro Nicola Candia, Cristian M. Malnero, Ismael Bermudez, Mauricio Vargas Lopez, Gregorio David Ríos, Cecilia Rotondaro, Nicasio Cuneo, David T. Curiel, Osvaldo L. Podhajcer, Maria Veronica Lopez

**Affiliations:** 1Laboratory of Molecular and Cellular Therapy, Instituto Leloir, IIBBA-CONICET, Avenida Patricias Argentinas 435, Ciudad Autónoma de Buenos Aires C1405BWE, Argentina; aalfano@leloir.org.ar (A.A.); ecafferata@leloir.org.ar (E.G.A.C.); mgangemi@leloir.org.ar (M.G.); alenicola90@gmail.com (A.N.C.); mvargas@leloir.org.ar (M.V.L.); grios@leloir.org.ar (G.D.R.); crotondaro@leloir.org.ar (C.R.); 2Facultad de Ingeniería, Universidad Argentina de la Empresa, Lima 775, Ciudad Autónoma de Buenos Aires C1073AAO, Argentina; cm.malnero@gmail.com (C.M.M.); ibermudez@uade.edu.ar (I.B.); 3Servicio de Ginecología, Departamento de Cirugía, Hospital Municipal de Oncología Maria Curie, Avenida Patricias Argentinas 750, Ciudad Autónoma de Buenos Aires C1405BWE, Argentina; cuneonic@hotmail.com; 4Division of Cancer Biology, Department of Radiation Oncology, School of Medicine, Washington University in St. Louis, St. Louis, MO 63110, USA; dcuriel@wustl.edu

**Keywords:** gynecologic neoplasms, cytokines, ovarian cancer, gene therapy, adenovirus

## Abstract

More than one million women are diagnosed annually worldwide with a gynecological cancer. Most gynecological cancers are diagnosed at a late stage, either because a lack of symptoms, such as in ovarian cancer or limited accessibility to primary prevention in low-resource countries, such as in cervical cancer. Here, we extend the studies of AR2011, a stroma-targeted and tumor microenvironment responsive oncolytic adenovirus (OAdV), whose replication is driven by a triple hybrid promoter. We show that AR2011 was able to replicate and lyse in vitro fresh explants obtained from human ovarian cancer, uterine cancer, and cervical cancer. AR2011 was also able to strongly inhibit the in vitro growth of ovarian malignant cells obtained from human ascites fluid. The virus could synergize in vitro with cisplatin even on ascites-derived cells obtained from patients heavily pretreated with neoadjuvant chemotherapy. AR2011(h404), a dual transcriptionally targeted derived virus armed with hCD40L and h41BBL under the regulation of the hTERT promoter, showed a strong efficacy in vivo both on subcutaneous and intraperitoneally established human ovarian cancer in nude mice. Preliminary studies in an immunocompetent murine tumor model showed that AR2011(m404) expressing the murine cytokines was able to induce an abscopal effect. The present studies suggest that AR2011(h404) is a likely candidate as a novel medicine for intraperitoneal disseminated ovarian cancer.

## 1. Introduction

Worldwide, more than one million women are diagnosed annually with a gynecological cancer. Non-specific symptoms (such as in ovary cancer) and disparities in accessibility to health services (such as in cervical cancer) explain the differences in gynecological cancer outcomes globally. Most gynecological cancer types are characterized by an accompanying stroma that not only supports malignant cell growth, but is also largely responsible for the high resistance of the cancer to conventional and targeted therapies [1,2].

Epithelial ovarian cancer (EOC) is one of the most common gynecological cancers, with one of the highest mortality rates in women (4.7% of the total number of deaths in women; excluding basal cell carcinoma) close to cervical and uterine cancer (7.7% of total number of deaths in women, excluding basal cell carcinoma) mortality rates [3,4]. Globally, EOC is the eighth most common malignancy diagnosed among women accounting for more than two hundred thousand deaths globally [4]. More than 300,000 women were estimated to have been diagnosed with EOC worldwide in 2020 [4]. Due to the late onset of symptoms and the absence of early screening and detection modalities, EOC is usually diagnosed at an advanced stage and approximately 75% of women present with advanced stage disease (stage III or IV) [5]. Whilst patients initially respond to chemotherapy, 80% of them rapidly develop chemoresistance, and the recurrence is pretty high [6,7]. Exfoliated individual cells or spheroids of ovarian cancer cells disseminate through the peritoneal cavity colonizing the mesenterium, the omentum and the diaphragm, and also the external layers of organs, such as intestine and spleen [8].

Ovarian cancer shows a remarkable resistance to available therapies. The current management of ovarian cancer (stage II–IV) at its presentation consists of cytoreductive surgery combined with platinum-based chemotherapy combined or not with targeted maintenance hormonal therapy [9]. However, almost 30% of patients have primary platinum resistance, 80% of patients will rapidly become refractory to the treatment, and almost all patients will ultimately succumb to their disease [7]. Interestingly, recent studies have shown that a higher stroma proportion at the tumor at the initial diagnosis of ovarian carcinoma is associated with eventual emergence of platinum chemoresistance [10]. With current five year survival rates under 50%, and 15% of women with ovarian cancer dying within a few months of diagnosis, there is an urgent need for novel treatments for this deadly disease.

Numerous treatment options are currently being evaluated in the setting of recurrent EOC. These include targeted therapies, such as the anti-VEGF antibody bevacizumab and poly (ADP ribose) polymerase inhibitor (PARPi) therapy, which have shown some efficacy in extending progression-free survival rates. However, it is important to note that these treatments have a disadvantage of not significantly extending the overall survival [11,12]. Bevacizumab, combined with platinum-based chemotherapy, has been recommended by the National Comprehensive Cancer Network (NCCN) guidelines as a first-line treatment for EOC [9]. This recommendation is based on its demonstrated advantage of a prolonging progression-free survival (PFS) that does not last more than 3.5 months in average [13]. Interestingly, the combination of the anti-PD1 and anti-CTLA4 check point inhibitors showed promising results in platinum-resistant ovarian cancer at the six month-interim analyses with an overall response rate (ORR) of 34% (doubling the results of nivolumab monotherapy) [14,15]. In November 2022, mirvetuximab soravtansine (a conjugated antibody targeting the folate receptor α to inhibit microtubules) was granted an accelerated approval by the FDA for the treatment of patients with folate receptor α positive, platinum-resistant EOC who have received 1–3 prior systemic treatment regimens; the median duration of response was less than 7 months [16]. The efficacy of the different and novel targeted therapies reduces with each recurrence and even the more advanced targeted medicines are still far from providing reliable therapeutics for this deadly disease.

Oncolytic viruses are a state of the art therapeutic strategy for cancer treatment. Oncolytic adenoviruses (OAdV) can be engineered with tumor specific promoters (TSP) and transcriptionally targeted to selectively attack and kill target cells [17,18]. In previous studies, we have shown that hybrid TSPs can be designed to target the cancer stromal cells compartment in addition to the malignant cell compartment [18]; moreover, TSPs can be engineered with the addition of tumor microenvironment responsive motifs [19]. In the present study we extend the studies of AR2011, a stroma-targeted, tumor microenvironment responsive OAdV, by showing its lytic capacity on fresh explants obtained from different gynecological cancers (ovarian, uterus, and cervical cancer). We also show its lytic capacity on human ovarian cancer cells obtained from peritoneal ascites combined with cisplatin. Finally, we describe the in vivo efficacy in different murine models of a AR2011-derived version armed with cytokines’ genes.

## 2. Results

### 2.1. AR2011 In Vitro and In Vivo Lytic Activity

#### 2.1.1. On Fresh Explants Obtained from Human Gynecological Cancers

AR2011 is a stroma-targeted, tumor microenvironment responsive, oncolytic adenovirus (OAdV) whose replication is driven by a triple hybrid promoter based on a 0.5 Kb selected fragment of the SPARC promoter combined with hypoxia and NFkB-responsive elements; a parental version of this OAdV showed a remarkable efficacy in preclinical models of ovarian cancer [18]. In the present studies, we initially aimed to establish if AR2011 lytic effect could be extended to other gynecological cancers beyond EOC. We initially aimed to establish that the triple hybrid promoter is indeed active in human cervical cancer cell lines. By using a non-replicative virus, we observed a 2- to 10-fold increase in promoter activity under hypoxia and TNFα (as an activator of the NFkB elements of the promoter [19]) in human cervical cancer cell lines (Figure 1A). Next, we assessed AR2011 lytic activity on human cervical cancer cell lines. AR2011 exerted a remarkable lytic effect on the three cell lines assayed (Figure 1B). Maximal lytic activity was attained at 10–100 MOI (Figure 1B). HeLa cells were the most responsive cell line both to hypoxia and to TNFα (Figure 1B).

In previous studies, we have shown that a former version of AR2011 was able to replicate and lyse fresh explants of human primary and metastatic ovarian cancer [18]. Here, we extended those studies by assessing AR2011 replication in additional gynecological human cancers, including cervical cancer fresh explants (Table 1), by quantifying adenoviral E4 levels as a surrogate marker of virion particles. AR2011 was able to replicate in 4/5 fresh explants of ovarian cancer (range of E4 ratio at 72 h vs. 5 h: 2.3–100.4), in 3/6 fresh explants of cervical cancer (range of E4 ratio 72 h vs. 5 h: 2.7–33.4), and in 4/5 fresh explants of uterus cancer (range of E4 ratio 72 h vs. 5 h: 2.6–120.5) (Figure 1C–F). AR2011 did not replicate at all in fresh explants of a normal uterus (Figure 1F). Although AdWT(F5/3) was able to replicate and lyse most gynecological cancer explants (Figure 1C–F), it also replicated and lysed fresh explants of two normal uteri (E4 ratio 72 h vs. 5 h: 4.0 and 29.0) (Figure 1F).

#### 2.1.2. On Malignant Ovarian Epithelial Cells Obtained from Ascites Fluid

More than 70% of women with ovarian cancer are diagnosed with advanced disease, and surgery combined with chemotherapy regimens (including platinum analogues and taxanes) are still the mainstay option in the frontline setting. In order to establish the potential use of AR2011 in combination treatment options for advanced ovarian cancer, we assessed its lytic capacity on malignant cells obtained from ascites fluid (OC-AF). In initial studies performed on three different samples of OC-AF, we selected an MOI of 100 that was sufficient to induce 50% inhibition of the malignant cells’ growth (Figure 2A). From previous data from the literature [20,21] and our own data, we selected 2.5 μg/mL, which is the IC50 of cisplatin on SKOV-3 that is accepted as a slightly resistant cell line to cisplatin. Fourteen OC-AF samples (3/14 with an IC50 below that of SKOV-3) were hence exposed to the combination of AR2011 at 100 MOI and cisplatin at 2.5 μg/mL compared to the exposure to each individual agent. We observed that the combination of AR2011 + cisplatin was able to kill malignant cells more efficiently than each single agent individually in all the samples (Figure 2B). Moreover, in 9/14 samples (64%), the combination was synergistic according to the Bliss independence model [22] used to analyze the drug combination data (Table 2).

#### 2.1.3. In Vivo Efficacy of the Combination of AR2011 and Cisplatin

Next, we tried to assess the efficacy of combining AR2011 with cisplatin in an in vivo model. In the absence of productive adenovirus replication in most murine organs and murine malignant cells, the in vivo efficacy of oncolytic adenoviruses is usually tested in immunocompromised nude mice xenografted with human malignant cells or tissue. Efforts to obtain patient-derived xenografts from the fresh explants failed due to an insufficient amount of tissue; we also failed in trying to obtain tumors in SCID mice using ascites-derived malignant cells as the starting material. Therefore, we conducted an in vivo study in nude mice to assess the efficacy of using AR2011 in combination with cisplatin on intraperitoneal-established SKOV3 human ovarian tumors. Following initial studies to confirm a 100% tumor take, mice harboring established tumors were treated with suboptimal doses of AR2011 alone (3 doses of 1 × 10^9^ v.p.) or combined with three administrations of an optimal dose (6 mg/Kg) or suboptimal dose (1.5 mg/Kg) of cisplatin [23]. Our findings show that the survival of mice treated with the suboptimal dose of either AR2011 or cisplatin was similar to that of the control animals (Figure 2C). Interestingly, a suboptimal dose of AR2011 combined with an optimal dose of cisplatin significantly increased mice survival compared to any other group, although it did not reach statistical significance with the group of mice treated with cisplatin alone (*p* = 0.1791 (Figure 2C).

### 2.2. In Vitro and In Vivo Studies of AR2011(404), a Novel Oncolytic Vector Engineered to Express Immunomodulatory Genes

#### 2.2.1. Vector Construction and In Vitro Studies

Following the initial attack of the oncolytic virus, it is expected that the immunomodulatory genes carried by the virus would help to expand the secondary antitumor immune response. Therefore, and as a further step to bring AR2011 closer to the clinic, we engineered AR2011 with the human versions of membrane bound hCD40L and h4-1BBL under the control of the human telomerase (hTERT) promoter to obtain AR2011(h404) (Figure 3A). By using a combination of flow cytometry and Western blots, we were able to confirm that the hTERT promoter was able to drive the expression of hCD40L and h4-1BBL in SKOV-3 human ovarian cancer cells and in A549 human lung cancer cells, both expressing endogenous 4-1BBL (Figure 3B,C). On the other hand, AR2011 induced a decrease in hCD40L and h4-1BBL expression in target cells likely due to its cells’ lytic effect. In addition, we observed that AR2011 and AR2011(h404) were able to kill SKOV-3 and OV4 human ovarian cancer cells, A549 human lung cancer cells, and CaCo 2 human colorectal cancer cells quite similarly, indicating that the expression of the cytokines did not hamper AR2011(h404) lytic capacity compared to the parental AR2011 OAdV (Figure 3D).

#### 2.2.2. In Vivo Studies in Nude Mice

We performed different experiments in nude mice to assess the in vivo efficacy of AR2011(h404). Following preliminary studies to confirm 100% tumor take, nude mice were injected with tumorigenic inoculum of human ovarian cancer cells SKOV-3 in the flank. Three intratumor administrations of AR2011 led to a marked inhibition of tumor growth in most of the animals compared to control mice injected with PBS (Figure 4A); notably, AR2011(h404) administration led to an even higher inhibition of tumor growth, and 7/10 mice showed no detectable tumors at the end of the experiment (Figure 4A). Treatment with each one of the OAdVs reached statistical significance compared to control PBS-treated mice (Figure 4A). Macroscopic visualization showed the complete disappearance of the tumor in most mice already 15 days after the first administration of AR2011(h404) (Appendix A). Thus, the expression of the cytokines did not hamper AR2011(h404) activity that exhibited a slightly increased efficacy that did not reach statistical significance compared to AR2011. We confirmed that the cytokines are expressed in vivo; indeed, a new set of nude mice were injected s.c. with a tumorigenic inoculum of SKOV-3 cells followed by a single intratumor injection of either PBS, AR2011, or AR2011(h404); 72 hr. later, tumors were removed for the assessment of the cytokines’ expression. Western blot studies of the removed tumors showed that only samples obtained from mice treated with AR2011(h404) showed an in vivo expression of hCD40L (Appendix A). On the contrary, we were unable to see an increased expression of h41BB-L in vivo since SKOV-3 cells showed endogenous expression of the cytokine; in fact, PBS-treated mice showed the largest expression of h41BB-L while OAdVs-treated mice showed a diminished expression of h4-1BBL, most likely due to the elimination of SKOV-3 cells due to OAdV in vivo lytic effect (Appendix A).

We next moved on to assess the in vivo efficacy of AR2011(h404) in avoiding peritoneal dissemination of malignant cells in nude mice xenografted with SKOV-3 ovarian cancer cells. Mice harboring established intraperitoneal tumors were treated i.p. with 5 × 10^10^ v.p. either of AR2011 or AR2011(h404) while control mice received PBS. OAdVs or PBS administration in the peritoneum was repeated at days three and five. At the end of the study, when all control mice showed a swollen belly, the mice were sacrificed following institutional guidelines. At necropsy, the mice were photographed, and all the intraperitoneal tumors were excised, photographed, and weighed. In addition to liquid ascites, all the PBS-treated control mice showed an extended intraperitoneal dissemination, including large metastatic nodules in liver, diaphragm, intestines, and peritoneum walls (Appendix A). None of the mice treated with AR2011 were tumor free, but 4/9 mice had tumors ≤0.1 g. In addition, 4/9 mice developed ascites; interestingly, 3/10 mice treated with AR2011(h404) were tumor-free, and an additional mouse had a tumor ≤ 0.1 g (Appendix A). Only 3/10 mice developed ascites. Overall, we observed a statistically significant inhibition of tumor growth both in AR2011—as in AR2011(h404)-treated mice (Figure 4B). In order to confirm that tumors were indeed targeted by the virus, we assessed E4 adenoviral gene levels as a surrogate marker of virion number. Assessment of E4 levels at the end of the experiment confirmed the presence of virions in the tumor mass of 3/9 mice treated with AR2011 and 4/7 mice treated with AR2011(h404) (Figure 4C). Interestingly, E4 levels were higher (albeit not statistically significant) in tumor samples obtained from mice treated with AR2011(h404) (Figure 4C).

#### 2.2.3. In Vivo Studies in Syngeneic Mice Models

As mentioned above, most if not all murine tumor models do not support adenoviral replication and, hence, limit the use of adenoviruses to assess the induction of a secondary immune response further to adenovirus infection/replication in malignant target cells. Despite that limitation, we tried to establish whether we can make use of a syngeneic model that might be useful to demonstrate if the arming cytokines are active in vivo. We constructed AR2011(m404) where the hTERT promoter drove the transcriptional activity of the murine versions of CD40L and 41BB-L to obtain AR2011(m404). The in vitro and in vivo effects of AR2011(m404) were compared with AR2011. To reduce the liver uptake [24], we exchanged the hexon protein in both OAdVs by engineering most of the hAdV3 hexon protein instead of the native one. We observed that neither AR2011(H3) nor AR2011(m404) were able to kill ID8 murine ovarian malignant cells in vitro up to an MOI of 50,000 (Figure 5A), most likely related to the failure of adenovirus proteins synthesis in these murine cells [25]. Following assessment of additional cell lines, we found that AR2011(H3) and AR2011(m404) were able to kill CT26 cells at a very similar level, starting from an MOI of 10,000 and reaching around 60% cell killing at MOI 50,000 (Figure 5B). The difference observed between both CT26 and ID8 was not related to the promoter activity as the triple hybrid promoter was active in both cell lines (Appendix A). 

Based on these data, we decided to perform two different in vivo studies with CT26 cells aiming to explore cytokines’ contribution to the antitumor viral effect in vivo. In the first approach, Balb/c mice were injected in both flanks with a tumorigenic inoculum of 5 × 10^5^ syngeneic CT26 cells; nine days later, when tumors reached an average volume of 100 mm^3^, mice were injected in the left flank i.t. with 7.5 × 10^10^ v.p. of either AR2011(H3), AR2011(m404), or PBS. Mice received additional injections in the left tumor with the same amount of viral particles or PBS at days three and five after the first administration (Figure 5C). Tumor size was followed in both flanks of each individual mouse; the experiment was ended when all the PBS-treated mice needed to be sacrificed due to an average tumor size of ~2000 mm^3^ that occurred after 19 days. Interestingly, an intratumor administration of AR2011(H3) had absolutely no effect on the tumors’ growth compared to PBS-treated mice neither in the injected left flank nor in the non-injected right flank (Figure 5C,D). Interestingly, AR2011(m404) treatment induced a statistically significant inhibition on tumor growth in the injected left flank compared either with AR2011(H3)- or PBS-treated mice (Figure 5C). Most interesting, AR2011(m404) treatment also induced a statistically significant inhibition of tumor growth in the non-injected right flank at day 18, although ultimately this difference waned at the end of the experiment (Figure 5D). Since AR2011(H3) had no effect at all, it is compelling to suggest that the inhibition of tumor growth observed in both flanks with AR2011(m404) is due to cytokines expression. 

In the next in vivo study, we injected Balb/c mice in the right flank with a tumorigenic inoculum of 5 × 10^5^ CT26 cells; when tumors reached an average volume of 100 mm^3^, mice were injected in the contralateral left flank with CT26 cells pre-infected overnight either with 30,000 MOI of AR2011(H3) (named CT-AR cells) or AR2011(m404) (named CT-AR404 cells). Control mice were injected in the contralateral flank with CT26 cells pre-treated with PBS (named CT-PBS). Contrary to the previous study, we observed no tumor growth in the left flank of mice injected either with CT-AR cells or with CT-AR404 cells while PBS pre-treated CT26 cells growth was unaffected. Mice were followed up to day 28 when all control mice (injected with CT-PBS cells) reached 2 cm^3^ tumor size in the right flank and were sacrificed following institutional guidelines. In close coincidence with the previous in vivo study, we observed no effect on the mice’s survival following the administration of CT-AR cells compared to CT-PBS-treated mice (Figure 5E). On the contrary, most of the mice treated contralaterally with CT-AR404 cells were still alive at the end of the experiment, suggesting that the cytokines’ expression was responsible for the observed effect (Figure 5E).

## 3. Discussion

The potential of ovarian cancer cells to disseminate and metastasize is governed essentially by the interaction between the malignant cells and the surrounding extracellular matrix. The outcome of oncogenic events in epithelial cells can be significantly modified by the nature of surrounding cancer-associated fibroblasts and endothelial cells. Stromal cells could be implicated in the acquisition of a chemoresistant phenotype. An extensive infiltrative pattern with desmoplasia is one of the major features favoring metastases [26]. In a recent study aimed to identify novel molecular subtypes of ovarian cancer by gene expression profiling, a poor prognosis subtype was defined by a reactive stroma gene expression signature, correlating with extensive desmoplasia [27]. In previous studies, we have shown the therapeutic efficacy of a novel stroma-targeted oncolytic adenovirus, AR2011, on preclinical models of ovarian cancer including fresh human explants. We extend the previous data by showing here that AR2011 was able to replicate and lyse fresh explants obtained from additional gynecologic cancers; moreover, it was also able to replicate and eliminate in vitro in combination with mainstay chemotherapeutic agents, ovarian cancer cells obtained from ascites of advanced stage disease. Moreover, AR2011 armed to express hCD40L and h4-1BBL showed an extended in vivo efficacy on human tumors established in the flanks of nude mice; the armed OAdV was able to arrest and even completely eliminate intraperitoneally disseminated human tumors. In preliminary studies, the armed OAdV expressing murine cytokines were able to exert a limited but significant abscopal effect in syngeneic murine models.

Chemotherapy resistance is the major limitation of current treatments for ovarian cancer. Since most patients are treated with neoadjuvant and adjuvant chemotherapy, the development of OAdV aimed at reversing resistance and sensitizing cells to chemotherapy agents can be a major advance in the field. Different in vitro and in vivo approaches have been used combining non-replicative viruses expressing sensitizing agents to chemotherapy compounds. As an example, it was shown that ovarian cancer cells can be sensitized in vitro and in vivo to cisplatin with the adenoviral expression of the manganese superoxide dismutase gene [28]. Previous studies have also shown that the myxoma virus can be combined in vitro and in preclinical models with chemotherapy agents to treat mice with syngeneic ovarian tumors [29]. It has also been shown that paclitaxel resistance can increase the oncolytic virus lytic effect through a mechanism that involves the upregulation of viral receptors [28]. Previous studies from our group showed that AR2011 can replicate and lyse fresh explants of solid metastases arising from human ovarian cancer heavily pretreated with chemotherapeutic agents [18]. Here, we extend the data by showing that AR2011 can synergize in vitro with cisplatin to eliminate ovarian cancer cells obtained from patients’ ascites, even in those cases where patients have been previously treated with neoadjuvant chemotherapy. The present data also suggest that suboptimal doses of AR2011 could eventually enhance the therapeutic efficacy of cisplatin in vivo.

A major limitation for adenovirus use in preclinical studies is the absence of syngeneic models that can recapitulate a human *scenario*, in particular the secondary immune response that follows the initial administration of an OAdV. In the absence of a full syngeneic ovarian cancer model in rodents [30], most of the studies with OAdV are being performed in nude mice with the limitation of the absence of the full immune response. The lack of cross reactivity of hCD40L with murine models posed an additional limitation to assessing the immune response associated with arming AR2011 with human CD40L. Interestingly, CD40-L has been shown to sensitize epithelial ovarian cancer cells to cisplatin treatment, clearly indicating that activation of the CD40 intracellular pathway in cancer cells can be of relevance beyond CD40 effect in establishing the adaptive immune response [31]. Despite the limitation with human CD40-L, it was interesting to note that AR2011(404) demonstrated a slightly higher in vivo efficacy than AR2011 in nude mice studies. We cannot rule out that this enhanced in vivo effect of AR2011(404) could be related to 4-1BBL expression. Although with markedly reduced affinity, it was shown that human 4-1BBL can bind the murine 4-1BBL receptor [32]. This cytokine has been shown to expand T cells and NK cells, and it is likely that NK activation could explain the slightly improved activity of AR2011(404) in vivo in the nude mice model. The activation of the 4-1BB pathway in T cells restoration of effector functions has been hampered by the liver toxicity induced by soluble agonists [33]. Therefore, it was compelling to express 4-1BBL locally under the regulation of the extremely specific hTERT promoter. Interestingly, the activation of 4-1BB with agonists has been used in combination with other immune check point agents, such as PD-1 and TIM-3, in murine models of ovarian cancer with remarkable efficacy [34].

Regarding the immune-mediated response induced by the armed AR2011(m404), we were disappointed by the fact that the parental and modified viruses were unable to kill ID8 cells in vitro. However, it was interesting to see in preliminary studies in a surrogate CT26 colorectal cancer model that the expression of murine CD40-L and 4-1BBL was able to induce not only a clear anti-tumor immune-mediated response locally, but also a modest but statistically significant abscopal effect, especially when mice harboring established tumors were treated with preinfected CT26 cells. Both the local and the abscopal effects were observed upon the administration of AR2011(m404) while AR2011(H3)-lacking cytokine expression was unable to induce any antitumor effect in this syngeneic model, neither local nor systemic. Although this study should be taken with caution since it is difficult to extrapolate from a subcutaneous colorectal cancer model to intraperitoneal ovarian cancer, the data clearly show that the cytokines are expressed in vivo and are biologically functional.

Diagnoses at a late stage of most ovarian cancers make it imperative to develop innovative therapeutic approaches to tackle the disease. Both RNA and DNA oncolytic viruses have been used in the few clinical studies in ovarian cancer; all the clinical studies did not proceed after phase I or are recruiting patients for phase I. Measles viruses expressing the carcinoembryonic antigen (CEA) were used in a phase I trial administered intraperitoneally in advanced stage cases [35]. The treatment was well-tolerated and resulted in dose-dependent biological activity in a cohort of heavily pre-treated recurrent ovarian cancer patients [35]. The vaccinia virus and Reovirus reolysin also reached phase I trials but definitive reports are still lacking [36]. Oncolytic adenoviruses have also been used in a few clinical trials in ovarian cancer with unclear effects. After initial studies with the 55K-E1B-deleted dl1520 oncolytic adenovirus was halted after phase I with no clear benefit [37], few other groups reached the phase I trial stage with modified oncolytic adenoviruses. Ad-delta24-RGD is an oncolytic adenovirus modified in the knob fiber domain to express an RGD moiety able to retarget the virus to integrins in a viral receptor-independent manner; this OAdV was well-tolerated in a phase I trial after intraperitoneal administration and showed promising clinical activity [38]. A variant of this OAdV expressing GM-CSF used in a compassionate mode in very few ovarian cancer patients was well-tolerated and appeared to induce an anti-tumor immune response [39]. The combination of this OAdV with a daily low dose of cyclophosphamide was also attempted in a phase I trial that involved few ovarian cancer patients with promising results [40]. Few other trials are still under patients’ recruitment [41]. We provide here novel data on the stroma-targeted AR2011 OAdV that has been shown to kill ovarian cancer cells obtained from liquid ascites corresponding to patients pre-treated or not with chemotherapy. AR2011 could be combined with mainstay chemotherapy and was able to eliminate cells that seem to be refractory to chemotherapy. Moreover, AR2011 could be armed with transcriptionally targeted cytokines to be expressed only in tissues overexpressing the hTERT gene. The hTERT proximal promoter has already been used to drive specific adenoviral replication in malignant tissues and oncolytic cell death [42]. OAdVs driven by hTERT have entered clinical trials stages in different cancer cell types.

## 4. Materials and Methods

### 4.1. Ethics Statement

All the experiments were approved by the Institutional Animal Care and Use Committee of the Fundación Instituto Leloir (Protocol #69OP, 2021). The Fundación Instituto Leloir has an approved Animal Welfare Assurance as a foreign institution with the Office of Laboratory Animal Welfare (NIH), Number F18-00411 (2023). All the patients at the Hospital Maria Curie signed an informed consent for samples’ use for research. All the samples reached Instituto Leloir in an anonymized code. The study was approved by the Ethics Committee of Hospital Maria Curie and by the Bioethics Committee of Fundación Instituto Leloir that also approved the use of human samples.

### 4.2. Cell Lines and Cell Culture

The human ovarian cancer cell lines SKOV-3 and OV-4, A549 lung carcinoma cells, human embryonic retinoblasts 911, and the murine colon carcinoma cells CT26 were already described [18]. The human cervical cancer cells (HeLa, CCL-2; SiHa, HTB-35 and Ca Ski, CRL-1550) were obtained from the ATCC (Manassas, VA, USA). HEK293 cells were purchased from Microbix (Toronto, ON, Canada). All the cell lines were grown in the recommended medium supplemented with 15% fetal bovine serum (Natocor, Cordoba, Argentina), 2 mM glutamine, 100 U/mL penicillin, and 100 μg/mL streptomycin and maintained in a 37 °C atmosphere containing 5% CO_2_.

### 4.3. Construction and Production of the Oncolytic Adenoviruses

The main features of AR2011 have already been described [43]. AR2011(404) was derived from AR2011 and expressed either human (h) or murine (m) CD40-L and 4-1BBL. For AR2011(h404) construction, a 2.8 Kb *Bgl*II fragment containing the hTERT promoter followed by the sequence encoding for hCD40L, an internal ribosome entry site (IRES) from encephalomyocarditis virus and h4-1BBL, was cloned into the *Bgl*II site of pshuttle 2kbHREF512ΔRb in the 3′–5′ orientation. A similar design was used for cloning a 2.9 Kb fragment in AR2011(m404) that included the murine version of each cytokine under hTERT. Both CD40L and 4-1BBL are the full length membrane bound forms and have a deletion in the cleavage site FEMQK (in hCD40L) and FEMQR (in mCD40L). All the DNA was synthetized at Genscript (Piscataway, NJ, USA) following our own design. All the constructs were confirmed with automatic sequencing at Macrogen (Seoul, Republic of Korea).

For the recombination steps, the pshuttle containing the sequences encoding for the human cytokines was linearized with *Pme*I and co-transformed with pvK500C F5/3 in BJ5183 cells to obtain the viral plasmid corresponding to AR2011(h404). To obtain AR2011(m404), the pshuttle encoding the sequences for the murine cytokines was recombined with pVK500C F5/3 where the entire hexon protein was replaced with the hexon protein of hAdV3. For hexon exchange, an adenovirus 5 backbone without hexon named pARΔHexon was prepared as described [44]. A 6.9 Kb fragment containing the entire hexon of hAdV3 and flanking regions from hexon 5 was released with *Sfi*I from pAd5H3/GL [44] and recombined with pVKΔHexon linearized with *Asis*SI. As a control of AR2011(m404), we constructed AR2011(H3) modified in the hexon protein following a similar procedure. The different recombined adenoviral genomes were linearized with *Pac*I and transfected in 911 cells. The rescued adenoviruses were used to infect HEK293 cells to produce the viral stocks [45]. All the constructs were confirmed with restriction pattern and automatic sequencing.

### 4.4. In Vitro Cytotoxicity Assay

For determination of virus-mediated cytotoxicity, 1 × 10^4^ cells were seeded in 24-well tissue culture plates and infected with the oncolytic adenoviruses at the indicated MOI [45]. Hypoxic and normoxic conditions as well as addition of TNFα to recreate an inflammatory environment was already described [19]. After 6 days, cell viability was measured using the CellTiter 96 AQueous One Solution Cell Proliferation Assay (MTS assay; Promega, Madison, WI, USA).

Fresh human explants were obtained at the Hospital Municipal de Oncología Marie Curie, Buenos Aires, Argentina, following institutional review board approval. Written informed consent was obtained from each patient. The declaration of Helsinki was followed in all the protocols. Samples were kept in RPMI medium on ice (Invitrogen, Carlsbad, CA). Time from harvest to slicing was kept at an absolute minimum (<2 h). Between 6 to 10 slices, 1–2 mm depth, were placed in 24-well plates followed by the addition of the virus at 500 MOI in 500 μL of DMEM/F12 including 2% *v*/*v* FBS, 1% antibiotics, and 1% L-glutamine [46]. Infections were allowed to proceed for 5 h where 3–5 slices were harvested for E4 quantification. In the remaining slices, the medium was replaced with fresh DMEM/F12 containing 10% FBS until the end of the experiment at 72 h. For assessment of E4 levels as a surrogate of viral particles, DNA was obtained from tissue slices, and qPCR for E4 was performed as described [18].

### 4.5. Isolation of Malignant Cells from Ovarian Cancer Liquid Ascites

Samples obtained from the Hospital Marie Curie were centrifuged at 1500 RPM for 10 min to clear the ascites from cells. Cells were incubated in DMEM/F12 supplemented with 15% of FBS, brought to confluence, and stored in liquid nitrogen until use. Each cells’ isolate was named as OC-AF followed by the respective number.

### 4.6. In Vitro Lytic Assays in Combination with Cisplatin

Malignant cells obtained from liquid ascites were seeded in 96-well tissue culture plates (5 × 10^3^ per well) and incubated for 48 h with cisplatin at a final concentration of 2.5 µg/mL for 48 h. After medium removal, cells were trypsinized, quantified, and replated in the presence of AR2011 at 100 MOI for another 96 h. When only cisplatin was used, cells were plated in the presence of cisplatin at 2.5 µg/mL for 48 h followed by medium addition without virus for another 96 h; for the control with virus alone, cells were plated only in medium for 48 h followed by incubation in the presence of AR2011 (MOI 100) for another 96 h as described above. At the end of the experiments cell viability was assessed with the MTS system.

### 4.7. Assessment of CD40L and 4-1BBL Expression

For assessment of hCD40-L expression by flow cytometry, A549 and SKOV-3 cells were infected with AR2011(h404). At the end, cells were harvested in 0.5 mM EDTA, washed, and resuspended in PBS containing 0.5% BSA at a concentration of 5 × 10^6^ cells/mL followed by incubation with phycoerythrin (PE)—conjugated hCD40L monoclonal antibody (eBioscience, San Diego, CA, USA). A PE-conjugated mouse IgG1 kappa isotype (clone P3.6.2.8.1, eBioscience, San Diego, CA, USA) was used as a matched control antibody (eBioscience, San Diego, CA, USA). The cells were washed again and resuspended in 0.4% paraformaldehyde in PBS prior to analysis with a FACS Calibur flow cytometer (Becton Dickinson, Oxford, UK). Ten thousand cells were analysed in each case.

For Western blot assessment of h4-1BBL in cell lines, SKOV3 and A549 cells were seeded at 1 × 10^5^/well in a 6 multiplate well plate. The next day cells were infected with AR2011(h404) and incubated at 37 °C for 30 h. Cells were then harvested, and total protein extracts were prepared in lysis buffer containing 10 mM Tris (pH 7.5), 1 mM EDTA, 150 mM NaCl, 1% Triton X-100, 0.5% deoxycholic acid, 0.1% SDS, and a protease inhibitor cocktail. Total protein extracts were separated in 10% SDS-PAGE and transferred to nitrocellulose membranes (Bio-Rad Laboratories, Hercules, CA, USA). The membranes were probed with anti-human 4-1BBL antibody (ab68185, Abcam, Waltham, MA, USA) and anti β-actin antibody (A4700; Sigma, St. Louis, MO, USA). HRP-Goat anti Rabbit (111035144, Jackson, West Grove, PA, USA) and HRP Goat anti Mouse (115035003: Jackson, NE, USA) were used as secondary antibodies. Enhanced chemiluminescence (ECL) reagents were used to detect the signals following the manufacturer’s instructions (Amersham, Piscataway, NJ, USA). Antibody signals were digitized by Image Quant LAS 4000 (GE-Cytiva, Marlborough, MA, USA). Anti β-actin antibody (A4700; Sigma, USA) and anti-αTubulin Ab (12g10 DSHB) were used as a loading control. HRP-Goat anti Rabbit (111035144: Jackson, NE, USA and HRP Goat anti Mouse (115035003: Jackson, NE, USA)) were used as secondary antibodies. Enhanced chemiluminescence reagents were used to detect the signals following manufacturer’s instructions (Amersham, USA). Antibody signals were digitized by Image Quant LAS 4000 (GE-Cytiva, MA, USA).

For assessment of hCD40-L and h4-1BBL expression in tumors, the tumor sample extracts were prepared using lysis buffer containing 50 mM Tris (pH 7.3), 150 mM NaCl, and 0.1% Tween 20 plus Halt protease inhibitor cocktail (8775, Thermo, Rockford, IL, USA). One hundred μg of protein extract was separated in 12% SDS-PAGE and transferred into nitrocellulose membranes. The membranes were probed with anti-h4-1BBL (ab68185, Abcam, MA, USA) and anti-hCD40L (ab2391, Abcam, Waltham, MA, USA). Anti β-actin antibody (A4700; Sigma, USA) and anti-α Tubulin (12g10 DSHB, Iowa City, IA, USA) were used as a loading control. HRP-Goat anti Rabbit (111035144, Jackson, NE, USA) and HRP Goat anti Mouse (115035003, Jackson, NE, USA) were used as secondary antibodies. Enhanced chemiluminescence reagents were used to detect the signals following manufacturer’s instructions (Amersham, USA). Antibody signals were digitized by Image Quant LAS 4000 (GE-Cytiva, MA, USA).

### 4.8. In Vivo Studies with Nude Mice

For combination studies with AR2011 and cisplatin, six- to eight-week-old female nude mice were obtained from the animal facility of the University of La Plata, Buenos Aires. After an acclimation period in Instituto Leloir animal facility, mice were injected with 6 × 10^6^ SKOV-3 cells intraperitoneally that we established as the inocula for 100% animal take. Mice were injected either with (a) PBS; (b) AR2011 at suboptimal dose (10^9^ vp/mouse/3 injections at days 8, 10, and 12); (c) cisplatin alone at optimal dose (6 mg/kg/mouse/3 injections at days 8, 10, and 12); (d) combination of cisplatin at the optimal dose injected at days 8, 10, and 12 (followed by AR2011 at suboptimal dose injected at days 14, 16, and 18); (e) combination of suboptimal dose of cisplatin (1.5 mg/kg/mouse/3 injections at days 8, 10, and 12); and (f) combination of a suboptimal dose of cisplatinum injected at days 8, 10, and 12 and AR2011 at a suboptimal dose injected at days 14, 16, and 18. Mice were sacrificed following institutional guidelines if they showed difficulty walking, cachexia, palpable i.p. tumors, or accumulation of ascitic fluid. The experiment was terminated at day 90 when it became clear that surviving animals were essentially cured.

For studies with subcutaneous tumors, mice were injected with 4.5 × 10^6^ SKOV3 cells in the flank. Tumor volumes were followed with caliper measurement every 2–3 days. Once tumors reached an average volume of 100 mm^3^, mice were injected intratumorally with 1 × 10^10^ v.p. of AR2011 or AR2011(h404) in 30 μL (as well as an equivalent volume of PBS for the control group). Viral or PBS administration was repeated 2 and 4 days later. Mice were followed with daily observations on general health until the control group (PBS) needed to be sacrificed due to animal distress following the approved protocol of the Institutional Animal Care and Use Committee of Instituto Leloir. At the end of the study, remaining tumors were excised and weighed. For the in vivo assessment of hCD40L and h4-1BBL expression, mice were injected with 4.5 × 10^6^ SKOV3 cells in the flank. When tumors reached 100 mm^3^ mice were administered i.t. with 5 × 10^10^ v.p. of AR2011, AR2011(h404), or PBS. Seventy-two hours later mice were sacrificed, the tumor area was removed, and a protein extract was prepared for Western blot analyses as described above. 

For the study in nude mice to compare AR2011 with AR2011(h404), we followed the procedure described above in essence. Briefly, mice were injected with 6 × 10^6^ SKOV-3 cells. Five days later, mice were injected i.p. with 5 × 10^10^ vp in 400 μL of AR2011, AR2011(h404), or PBS. Injection was repeated 3 and 5 days later. Three mice per group were sacrificed 3 days after the last viral administration for assessment of E4 levels as a surrogate marker of viral particles [18]. Mice were followed as described above. The remaining mice were sacrificed at day 54 of the first viral administration following institutional guidelines described above and remaining i.p. tumors were photographed in situ, excised, weighed, and a sample used for assessment of E4 levels.

### 4.9. In Vivo Studies with Syngeneic Models

Balb/c mice (6–8-week-old male) were obtained from the animal facility of the University of La Plata. After an acclimation period in Instituto Leloir animal facility, mice were injected in both flanks with 5 × 10^5^ syngeneic CT26 colorectal carcinoma cells. The tumorigenic inoculum (with 100% tumor take) was selected from a pilot study where mice were injected with 3 × 10^5^, 5 × 10^5^, and 1 × 10^6^ cells in 100 μL of PBS. Once tumors reached a volume of 75–100 mm^3^, mice were injected in the left tumor with 7.5 × 10^10^ vp of either AR2011(H3) or AR2011(m404) in a final volume of 50 μL PBS. Control mice were injected with 50 μL PBS. Tumors were measured bi-weekly in two dimensions with a caliper. The mice were followed until they need to be sacrificed due to a tumor size that exceeded 2000 mm^3^. In a second type of experiment, Balb/c mice were injected in the left flank with 5 × 10^5^ CT26 cells. Once tumors reached an average volume of 100 mm^3^, mice were injected in the contralateral right flank with 50 μL containing 5 × 10^5^ CT26 cells pre-infected overnight with 3 × 10^4^ MOI of either AR2011(H3) or AR2011(m404). Control CT26 cells were pretreated with PBS. Mice tumors were assessed with digital calipers, and the volume was obtained with the following formula: volume = 0.52 × (width)^2^ × length.

### 4.10. Statistical Analysis

For Figure 1B–E, Figure 2B, Figure 4A–C and Figure 5C,D, the statistical difference between groups was determined by a *t*-test and the F test (H. J. Motulsky, GraphPad Statistics Guide. http://www.graphpad.com/guides/prism/7/statistics/index.htm, accessed on 5 March 2016). The survival curve in Figure 5E was analyzed with Log-rank (Mantel-Cox) test. A *p*-value of <0.05 was considered statistically significant. Data analysis was performed with the GraphPad Prism 8.0 (GraphPad Software, Inc., San Diego, CA, USA). Bliss independence model [22] was used to analyze drug combination data. The Bliss method compared the observed combination response (Y(O)) with the predicted combination response (Y(P)), which was obtained based on the assumption that there was no effect from drug–drug interactions. Typically, the combination effect is declared synergistic if Y(O) is greater than Y(P).

### 4.11. Patents

Part of the present data is included in the US patent application No.: 16/797,291 whose inventors are D.T.C., O.L.P. and M.V.L.

## 5. Conclusions

Based on previous evidence and the current study, it is highly likely that AR2011(404) will provide high lytic selectivity and cytokine expression specificity due to the combination of the triple hybrid SPARC-based promoter and the hTERT promoter. The fact that the virus expresses the pseudotyped fiber 5/3 also provides higher selectivity in terms of ovarian cancer cell targeting. The evidence that our virus showed efficacy in samples from additional gynecologic cancers makes this virus a reliable candidate to reach clinical trials.

## Figures and Tables

**Figure 1 ijms-24-09992-f001:**
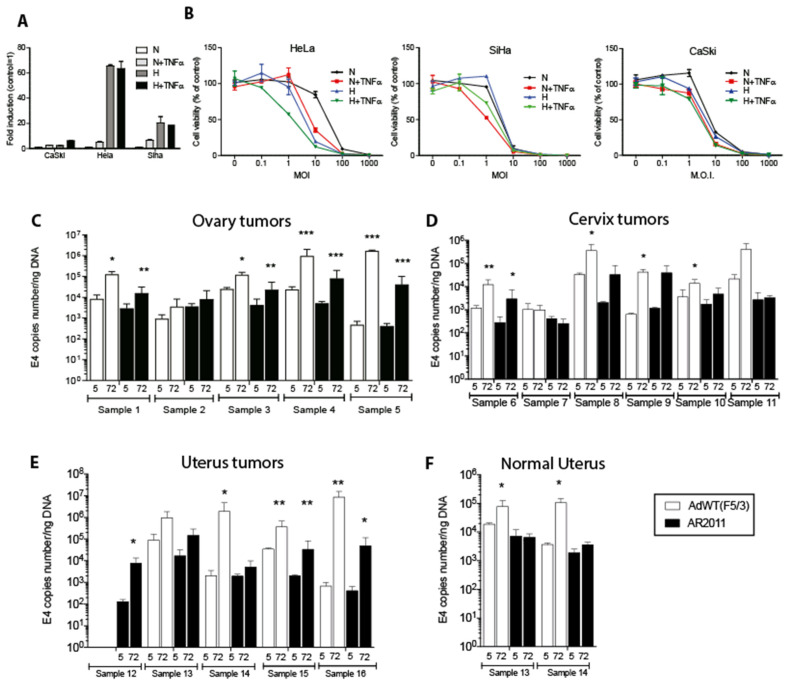
In vitro lytic activity of AR2011. (**A**) transcriptional activity of the triple hybrid promoter under normoxia (N), hypoxia (H), and TNFα as an inducer of NFkB translocation to activate NFkB-responsive elements located in the Promoter. (**B**) AR2011 lytic effect on different human cervical cancer cell lines. (**C**–**F**) AR2011 replication on fresh explants obtained from human ovarian, cervix, and uterus cancer samples and normal uterus. For further details and statistical analysis, see materials and methods. Statistical significance, * *p* < 0.05, ** *p* < 0.01, *** *p* < 0.001.

**Figure 2 ijms-24-09992-f002:**
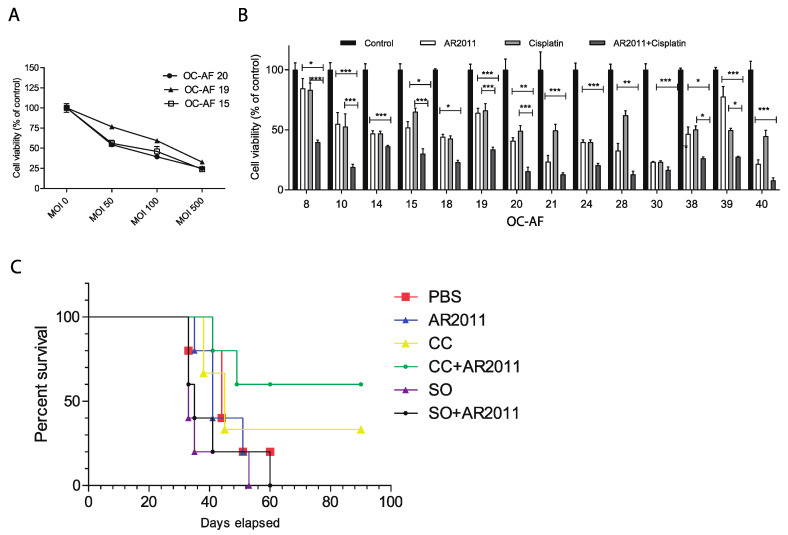
In vitro lytic activity and in vivo efficacy of AR2011. (**A**) Lytic effect of AR2011 at different MOIs on three different samples of ovarian cancer-ascitic fluids (OC-AF). (**B**) Lytic effect of AR2011 at MOI 100 on different samples of OC-AF combined or not with cisplatin. (**C**) Kaplan–Meier survival curve of nude mice harboring 8-day-old intraperitoneal SKOV-3 tumors. Mice were i.p. injected either with PBS, AR2011, cisplatin, or combinations. ODC (optimal dose of cisplatin), SDC (suboptimal dose of cisplatin). For further details, see Materials and Methods. * *p* < 0.05, ** *p* < 0.01, *** *p* < 0.001.

**Figure 3 ijms-24-09992-f003:**
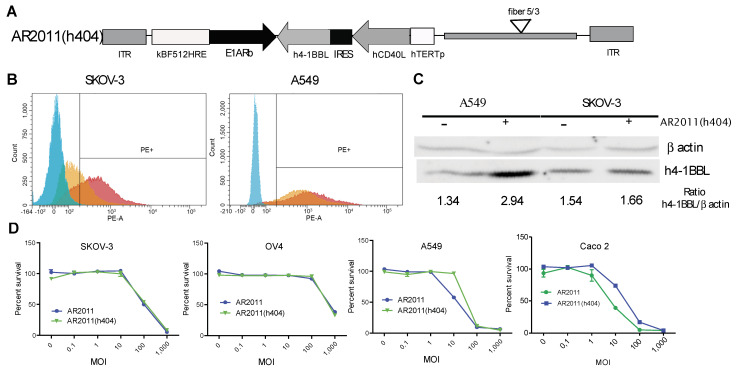
Cytokines expression and lytic activity of AR2011(h404). (**A**) Scheme of AR2011(h404) genome structure showing that the cytokines cassette under hTERT regulation was cloned in the 3′–5′ orientation in the AR2011(h404) genome. (**B**) Cell surface expression of hCD40L in SKOV-3 (left) and A549 (right) cells: Light blue: Isotype control, Orange: AR2011(h404) MOI 100; Pink: AR2011(h404) MOI 1000. (**C**) Western blot for the detection of h4-1BBL expression. The blot shows that both cell lines expressed endogenous 4-1BBL. Data for cytokine expression was obtained 30 h after infection since after 48 h cells were completely lysed by the OAdVs. (**D**) In vitro lytic activity of AR2011(h404) in different human malignant cell lines. Percent of surviving cells was assessed with the MTS assay. AR2011 was used as a comparator.

**Figure 4 ijms-24-09992-f004:**
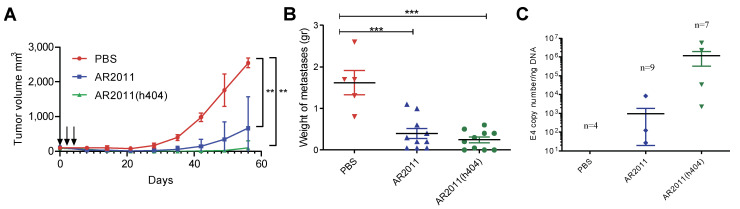
In vivo efficacy of the OAdVs. (**A**) Nude mice harboring established SKOV-3 tumors were injected with PBS, AR2011, or AR2011(h404) at the indicated times (day 0, 2 and 4, see arrows). Tumor growth was followed with digital calipers. At day 56, the experiment was finalized. (**B**) Mice harboring intraperitoneal 5-day-old SKOV-3 tumors were i.p. injected either with PBS, AR2011, or AR2011(h404) as described in the methods section. At the end of the study, tumors were removed, photographed, and weighted. (**C**) Levels of viral E4 in tumor samples as a surrogate marker of virion numbers. ** *p* < 0.01, *** *p* < 0.001. For statistical analysis, see materials and methods.

**Figure 5 ijms-24-09992-f005:**
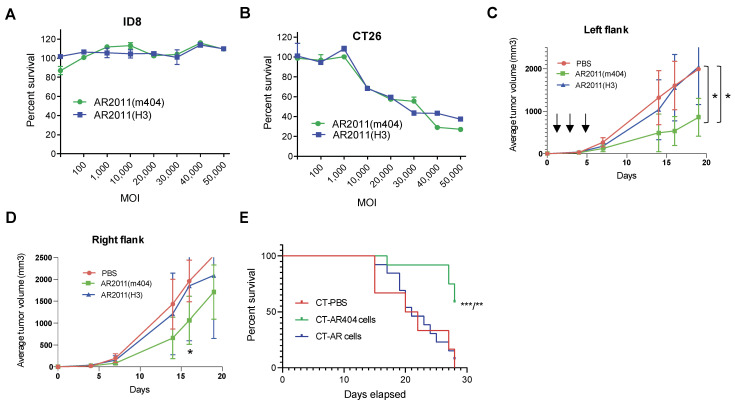
In vivo studies in syngeneic mice models. (**A**,**B**) In vitro lytic activity of AR2011(H3) and AR2011(m404) on ID8 and CT26 cells. Surviving cells were assessed with the MTS assay. (**C**,**D**) Mice harboring established CT26 tumors in both flanks were injected in the left flank either with PBS, AR2011(H3), or AR2011(m404) and followed as described in the text. The arrows indicate the days of AdOV or PBS injection (day 0, 3 and 5) * *p* < 0.05 (**E**) Kaplan–Meier survival curve of mice harbouring established CT26 tumors in the left flank injected in the right flank either with CT-PBS, CT-AR, or CT-AR404 cells. Statistical analysis: comparison of CT-PBS vs. CT-AR404, *p* = 0.0017 (**) and CT-AR vs. CT-AR404, *p* = 0.0009 (***) using the Long-Rank (Mantel-Cox) test. For further details, see materials and methods.

**Table 1 ijms-24-09992-t001:** Characteristics of the human fresh explants.

Sample	Pathology	Observation
1	Epithelial ovarian cancer	First cytoreduction
2	Epithelial ovarian cancer	Relapse
3	Epithelial ovarian cancer	Bilateral tumor Neoadjuvant chemotherapy C, P and Bevacizumab
4	Epithelial ovarian cancer	Neoadjuvant chemotherapy
5	Krukenberg tumor	First cytoreduction
6	Cervical cancer	Conization
7	Cervical cancer	Conization
8	Cervical cancer	Simple hysterectomy
9	Cervical cancer	Simple hysterectomy
10	Cervical cancer	Conization
11	Cervical cancer	Simple hysterectomy
12	Uterus cancer	Radical Hysterectomy
13	Uterus cancer	Normal uterus tissue after radical hysterectomy *
14	Uterus cancer	Normal uterus tissue obtained after radical hysterectomy *
15	Uterus cancer	Hysterectomy
16	Uterus cancer	Hysterectomy

* Fresh explants from histologically confirmed normal uterus were obtained from patients undergoing surgery due to uterus cancer.

**Table 2 ijms-24-09992-t002:** Analysis of the synergistic interaction of AR2011 and cisplatin combination on OC-AF samples.

OC-AF	Y(P) ^1^	Y(O) ^1^
8	0.30	**0.60**
10	0.71	**0.81**
14	0.66	0.64
15	0.66	**0.70**
18	0.81	0.77
19	0.57	**0.66**
20	0.80	**0.85**
21	0.88	0.87
24	0.78	**0.80**
28	0.80	**0.87**
30	0.88	0.83
38	0.77	0.74
39	0.61	**0.72**
40	0.90	**0.92**

^1^ Synergystic effect was obtained with the Bliss’s method. Synergistic cases are labelled in bold. For further details and statistical analysis, see materials and methods.

## Data Availability

Data is contained within the article or Appendix A, raw data are available upon request.

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
