# Peer review of "In Vitro and In Vivo Efficacy of a Stroma-Targeted, Tumor Microenvironment Responsive Oncolytic Adenovirus in Different Preclinical Models of Cancer"

_ijms, 2023, doi:10.3390/ijms24129992_

Round 1

Reviewer 1 Report

Review

The manuscript, “In vitro and in vivo efficacy of a stroma targeted, tumor microenvironment responsive oncolytic adenovirus in different preclinical models of cancer” by Alfano et al. extended the studies of AR2011, a stroma-targeted and tumor microenvironment responsive oncolytic adenovirus (OAdV) whose replication is driven by a triple hybrid promoter. It showed that AR2011 was able to replicate and lyse in vitro fresh explants obtained from human ovarian cancer, uterine cancer, and cervical cancer. The studies suggest that AR2011(h404) is a likely candidate as a novel medicine for intraperitoneal disseminated ovarian cancer.

Comments and Suggestions for Authors:

1- Please correct the keywords according to MeSH.

2- The section "4.10. Ethics statement" should be included at the beginning of the material and method section.

3- Please provide more detailed statistics about "Epithelial ovarian cancer (EOC) is one of the most common gynecological cancers, with one of the highest mortality rates in women close to cervical and uterine cancer mortality rates" in lines 44-45.

4-In lines 66-81, please state the advantages and disadvantages of each treatment method and mention the most important treatment.

5-What do you mean by "The potential of ovarian cancer cells to disseminate and metastasize into the perito-301 neal cavity is governed by, among others, the ECM composition."?

Please explain more about this.

6-Please state in lines 321-323 how many percent of chemotherapy patients face limitations and resistance

.

7- In line 24, remove the dot in the middle of the sentence

Reviewer 2 Report

In this paper, the authors describe an oncolytic adenovirus construct that is transcriptionally targeted to tumour cells.  Here they present a previously discussed oncolytic AdV, AR2001 that has a triple hybrid promoter consisting of a SPARC fragment, hypoxia response element and NFkB response element.  The authors are extending previous work in gynaecological cancers, by investigating the effects of the virus on a panel of gynaecological cancers.

This paper is of interest, but the aims of this paper seem to be a bit confused and the results not fully addressed at any stage.

The authors purport to address the efficacy of AR2001 in a panel of gynaecological cancer in vitro and show cytotoxicity and sensitization to chemotherapy in the samples they test.  This seems sound and expands the potential application for the virus.

However, the authors then move to different in vivo models, now looking at the effects of arming the virus in vivo.

They first use a nude model of SKOV-3 cells, which is more relevant to the aim of assessing gynaecological cancers, and show that there was an effect of both unarmed and armed viruses. 

11.       Why were SKOV3 chosen as opposed to using the clinical material that they had access to – this would make it more relevant to the overall aim of the paper.

22.       Why was sensitisation with chemotherapy not investigated given the in vitro results presented?

They also suggest that the armed virus improves outcomes in the nude xenograph model, but the mechanism, although proposed in the discussion is not investigated.

33.       If the authors are now wanting to address an arming strategy for their virus, they need to provide more data on the effect of arming the virus.  Actually, figure 4 shows that there is a potential for altered replication after arming the virus, so enhanced efficacy could be related to this?  Replication data (in vitro and in vivo) and immune landscape data (systemic, tumoural) are required to dissect the importance of arming.

In an immunocompetent model, CT26 cells are used which are colorectal cancer cell lines.  However, it is understandable as it can be difficult to find supportive murine cell lines for adenovirus experiments.  In this regard, the use of the CT26 is nice way to try and get around this issue.

44.       However, again the purpose was to investigate an effect of arming the virus and the authors went no way towards investigating how the arming of the virus affected the immune responses in the mouse (systemic and local). 

55.       Again, this model could also be used to investigate synergy with chemotherapy.

66.       As discussed, mice are a limiting model for investigation of AdV effects – why have the authors not considered more relevant models (hamsters) that are now widely accepted as supportive of AdV replication and more reflective of the human immune system.

Other comments

17.       These viruses are SPARC restricted, but no information or data is presented on SPARC expression in the tumor tissue samples presented.  This is important to determine the mechanism of selectivity in these models.

28.       How about the off-target effects/off target expression of SPARC  - is the murine model relevant here (i.e. mouse vs human SPARC activation of the promoter).  If not these limitations should be discussed.

39.       Figure 1:  Please clarify H=hypoxia; N=normoxia on the figure legend and the purpose of TGFB addition (NFkB activation)

410.       Why are E4 levels used to look at replication rather than traditional TCID50, which is much more suitable for determining levels of infectious virus production.

511.       Figure 4: In nude mice, the authors note that treatment with the armed virus lead to greater inhibition of tumour growth, but this is not backed up by the data shown (no significance) so re-think that statement in the first paragraph of 2.2.2.

612.       Figure 5: Given that AR2011 H3 can infect and kill cells in vitro, can the authors suggest why no in vivo effect was noted, despite 3 injections and retargeting of the virus, whereas in nude models, differences after using unarmed viruses are shown.

Round 2

Reviewer 1 Report

I do not have any more suggestion.

Reviewer 2 Report

The authors have appropriately responded to my comments and have updated the paper accordingly to reflect my concerns.  This is a much improved version.